# Farmer and Farm-Level Predictors of Pain Perception and Management During Routine Pig Husbandry Practices in Eastern Cape, South Africa

**DOI:** 10.3390/ani15243508

**Published:** 2025-12-05

**Authors:** Asemahle Tyutwana, Ziyanda Mpetile, Olusegun Oyebade Ikusika, Oluwakamisi Festus Akinmoladun

**Affiliations:** 1Department of Animal and Pasture Science, Faculty of Science and Agriculture, University of Fort Hare, Private Bag X1314, Alice 5700, Eastern Cape, South Africa; 201917527@ufh.ac.za (A.T.); zmpetile@ufh.ac.za (Z.M.); oikusika@ufh.ac.za (O.O.I.); 2Department of Animal and Environmental Biology, Faculty of Science, Adekunle Ajasin University, PMB 001, Akungba-Akoko 342111, Ondo-State, Nigeria

**Keywords:** animal welfare, pain management, pig husbandry, chi-square analysis, logistic regression, farmer perception

## Abstract

This study investigated how farmers in the Eastern Cape, South Africa, understand and manage pain during routine husbandry practices. Using surveys from 216 farmers, the study found that education, age, and farming experience significantly influenced knowledge and ability to assess animal pain. Painful procedures such as castration, tail docking, and teeth clipping were more likely to involve pain relief when done by trained personnel. Younger farmers and larger farms showed better awareness and management of pain. The study recommends farmer training, improved veterinary support, and standard welfare guidelines to promote humane pig production.

## 1. Introduction

Pig production plays a vital role in global agriculture, serving as a key source of animal protein and contributing significantly to food security, rural livelihoods, and national economies [1]. Globally, pork accounts for over 25% of total meat consumption, underscoring its importance in meeting the rising demand for animal-source foods driven by population growth, urbanization, and shifting dietary preferences [2,3]. However, in many developing countries, the potential of pig production is constrained not only by limited resources and infrastructure but also by poor animal welfare practices arising from inadequate knowledge, cultural norms, lack of enforcement of legislation, and insufficient access to veterinary services [1,4].

Routine husbandry procedures such as castration, tail docking, teeth clipping, and animal identification are widely practiced in pig farming to enhance productivity, prevent injuries, and manage aggressive behavior. While these interventions can be beneficial for herd management, they are known to cause varying degrees of pain, distress, and physiological stress responses if performed without appropriate analgesia, and they may also serve as potential entry points for pathogens through open wounds [5,6]. Pain assessment in farm animals has largely focused on changes in behaviour, physiology, and production or bodily functions [6]. For instance, castrated lambs displayed reductions in play behaviour, feed intake, and overall activity levels [7,8]. According to Bonastre et al. [9], the physiological response of piglets to castration manifests in increased glucose and cortisol concentration as well as a drop in skin temperature, independent of anesthesia or analgesia. Scientific evidence links poorly managed procedural pain to reduced growth rates, compromised immune function, and lower meat quality, indicating that effective pain management is not only an ethical imperative but also a key determinant of productivity and profitability [10]. Yet, in many low-resource farming systems, pain in pigs is often under-recognized and undertreated due to low awareness, cost-related barriers, and limited veterinary oversight [10]. These limitations are particularly pronounced in developing regions such as Sub-Saharan Africa, where livestock production systems face complex socioeconomic and infrastructural challenges that may indirectly affect animal welfare and pain management practices.

Within Sub-Saharan Africa, pig production generally faces overlapping socioeconomic and technical constraints that may indirectly influence welfare practices. The smallholder pig sector in countries such as Mozambique is severely challenged by endemic diseases, particularly African swine fever (ASF) and *Taenia solium* cysticercosis, as well as poor husbandry skills, limited market access, and widespread poverty [11,12,13,14]. Similar to other Southern African systems, Mozambican smallholders also contend with insufficient feed resources, absence of veterinary and breeding services, and lack of adequate housing or confinement [15,16]. In South Africa, Madzimure et al. [17] highlighted the productive potential of indigenous pig breeds, but numerous studies have reported that factors such as genetics, biosecurity, feed quality, housing, marketing limitations, environmental stressors, and land-tenure constraints remain major impediments to efficient production [18,19]. Comparable management deficiencies have also been described in Kenya, where smallholder farmers in Busia and Kakamega districts demonstrated limited knowledge of husbandry and zoonotic disease prevention [20], reflecting broader gaps in animal-health literacy and resource access across the region. However, despite the abundance of research on production constraints, disease management, and livelihood challenges, literature focusing specifically on farmers’ or farm workers’ perception of animal pain and welfare in Sub-Saharan Africa remains extremely sparse. This lack of empirical evidence underscores the need for regionally grounded studies that explore how cultural, educational, and structural factors shape pain perception and management practices in local pig production systems. Collectively, these challenges suggest that pig welfare, including pain management, may be further compromised by structural limitations common to smallholder production systems in Sub-Saharan Africa.

Pain management in pigs also presents legal, ethical, and logistical challenges. On one hand, farmers face increasing societal and regulatory pressure to improve animal welfare; on the other, concerns about the cost, efficacy, and residue risks of analgesic drugs may discourage their use [10]. Importantly, farmers’ values, knowledge, and perceptions play a central role in how animals are treated. Numerous studies have shown that knowledge, attitudes, and practices (KAP) related to animal welfare are shaped by a combination of socio-demographic (e.g., age, gender, education) and farm-level factors (e.g., herd size, production system, access to extension) [21,22,23]. Farmers who are more educated or professionally trained are typically more capable of recognizing pain symptoms, implementing timely interventions, and seeking veterinary support.

In the Eastern Cape Province of South Africa, pig production remains a growing agricultural enterprise with a mixture of smallholder and intensive systems. Despite the increasing interest in pork production, there is a limited understanding of how farmers and on-farm workers perceive pain associated with routine husbandry procedures and the extent to which pain management practices are influenced by farmer- and farm-level characteristics. This study, therefore, aims to assess the predictors of pain perception and management during routine husbandry procedures, focusing on knowledge, attitudes, practices, and socio-structural determinants that inform welfare-related decision-making.

## 2. Methodology

### 2.1. Study Area and Ethical Clearance

The study area was the Raymond Mhlaba Local Municipality situated in Amathole district in the Eastern Cape province of South Africa. This is situated in the former Ciskei where livestock farming is mainly practiced by the people in rural areas where most people practice subsistence farming. The study was cleared by the Inter- Faculty Human Research Ethics Committee (IFHREC) (Ref No. “MPE041STYU01”; 4 June 2024).

### 2.2. Study Design, Population and Sampling Technique

This study adopted a cross-sectional survey design to assess the predictors of pain perception and management during routine husbandry procedures in the Eastern Cape Province of South Africa. The primary objective was to examine the knowledge, attitudes, practices, and socio-structural determinants influencing welfare-related decision-making by pig farmers and on-farm workers.

The target population comprised both commercial (private or public) and smallholder pig farms operating in the Eastern Cape Province. Eligibility criteria for respondents included a minimum of 4 years’ experience in pig farming or on-farm work to ensure adequate exposure to and familiarity with pig behaviour and to ascertain active involvement in daily pig husbandry activities.

A purposive, multistage sampling strategy was employed: This include firstly, the identification of areas with significant pig production activity followed by selection of farms within each identified area using a combination of proximity to research base, production scale, referrals from extension officers, and farm registration lists and the inclusion of all willing and available respondents (farm owners and workers) present on each selected farm during the data collection period. Prior to data collection, all participants were provided with clear information regarding the purpose of the study, the voluntary nature of their participation, and the confidentiality of their responses. Written (or verbal, if appropriate) informed consent was obtained from each respondent before administering the questionnaire. Participants were assured that they could withdraw from the study at any point without any negative consequences.

### 2.3. Sample Size Justification

Sample size was calculated (216 respondents) using Cochran’s formula for cross-sectional studies:$${n_{0}}_{\;=\;\frac{Z^{2}.\left(1\;-\;p\right)}{e^{2}}}$$
where Z = 1.96 for a 95% confidence level, *p* = 0.5 (maximum variability, as no prior estimate exists and e = 0.07 (margin of error).

### 2.4. Data Collection and Procedure

A structured questionnaire was developed to capture (i) sociodemographic characteristics (age, gender, religion, education, farming experience, piggery size, type of piggery and access to technical advice), (ii) routine husbandry procedures (animal identification, tail docking, castration, teeth clipping, vaccination), including method, timing, reasons, responsible personnel, and use of analgesics, (iii) knowledge of animal welfare and behavioural observation in pigs and (iv) perceived pain intensity (0–10 numerical rating scale) for specified procedures.

The questionnaire was adapted from validated animal welfare perception instruments used in similar livestock research ensuring content validity [24]. It was translated into IsiXhosa and back-translated to English to maintain semantic equivalence. Pre-testing was conducted with 10 farmers outside the main sample to confirm clarity, cultural appropriateness, and relevance of questions. Minor adjustments were made to improve understanding. Reliability was assessed using Cronbach’s alpha for multi-item perception scales, with α ≥ 0.7 considered acceptable.

Data were collected using a combination of face-to-face interviews (for farmers with limited literacy) and self-administered questionnaires (for literate respondents). The interviews were conducted by trained enumerators familiar with animal welfare concepts, ensuring consistency in question delivery and minimising interviewer bias. Each interview lasted approximately 25–40 min. Responses were recorded directly onto printed questionnaires.

### 2.5. Statistical Analysis

Data was entered and cleaned in Microsoft Excel and analysed in IBM SPSS Statistics version 20. Descriptive statistics were computed to summarise farmer demographics, farm characteristics and pain management practices. Given that perceived pain scores were ordinal and non-normally distributed, non-parametric tests were applied: the Mann–Whitney U test was used to compare scores between two independent groups (e.g., male vs. female), while the Kruskal–Wallis H test compared scores across more than two categories (e.g., education level, age groups). For Kruskal–Wallis tests yielding significant results (*p* < 0.05), post hoc pairwise comparisons with Bonferroni correction identified specific group differences. Associations between perceived pain scores and continuous or ordinal predictors (knowledge, awareness, behavioural observation ability, and attitudes) were examined using Spearman’s rank correlation coefficients (ρ). To identify predictors of self-rated ability to assess pain in pigs, ordinal logistic regression (PLUM procedure, logit link function) was employed with sociodemographic and farm-level variables (age, gender, highest qualification, years of experience, piggery size, type of piggery, and awareness of animal welfare) as predictors; the proportional odds assumption was tested using the Test of Parallel Lines, and results were expressed as odds ratios (Exp(B)) with 95% confidence intervals. A *p*-value < 0.05 was considered statistically significant for all analyses. The ordinal logistic regression model is specified as:$$\log\left(\frac{P\left(Y\leq j\right)}{P\left(Y>j\right)}\right)=\emptyset j+\left(\beta 1X1+\;\beta 2X2\ldots .+\beta nXn\right)$$

For j = 1,2,3,4 thresholds between outcome categories, where ∅j = intercept (threshold) for category j.

## 3. Results

### 3.1. Sociodemographic Characteristics and Perception of Pain Management

The sociodemographic characteristics of the respondents are shown in Table 1. Most of the respondents (>80%) were above the age of 25 years and were largely male (74%). Most of the respondents (>80%) have a minimum of high school qualification and are either employed (44.4%) or engaged in private business (45.4%). Over 70% of the respondents have a minimum rating of ‘good’ in terms of their ability to assess pain in pigs as well as their knowledge of pain control in pigs.

The perception of respondents about pain management during routine husbandry procedures is shown in Table 2. Most of the respondents either disagree (49.5%) or strongly disagree (20.9%) that it is difficult to recognize pain in farm animals. While more than 50% of the respondents either ‘agree’ or ‘strongly agree’ that farm animals benefit from pain alleviation based on quick recovery rate (>80%) after routine husbandry procedure, the costly nature of most pain-relieving drugs (>50%) seems to deter their regular use.

### 3.2. Chi-Square Association Analysis of Selected Variables

The chi-square associations between socio-demographics and knowledge/practices (section A), routine husbandry procedures and pain control practices (section B), pig behaviour awareness and socio-demographics (section C) are shown in Table 3. In section A, the educational level of respondents was significantly associated with their ability to identify pain (*p* = 0.001), assess pain (*p* = 0.038), rate pain in pigs (*p* = 0.000) and their knowledge of the importance of animal welfare (*p* = 0.014). Also, the age of respondents was significantly associated with their ability to assess pain (*p* = 0.049) and the knowledge of behavioural expression in pigs (*p* = 0.035). Lastly, the years of experience of pig farming by the respondents were significantly associated with their ability to assess pain in pigs (*p* = 0.039). In section B, the procedure (castration, teeth clipping, tail docking and identification) performed is significantly (*p* < 0.05) associated with the use of analgesics. The type of person responsible for the procedure was significantly associated (*p* < 0.000) with the use of analgesics for castration, teeth clipping and tail docking. In section C, the education level of the respondents was significantly (*p* = 0.024) associated with the behavioural observation during routine husbandry procedure. The years of experience of the respondents were significantly associated with the perceived pain intensity for teeth clipping (*p* < 0.048) and the ability to identify (*p* < 0.016).

### 3.3. Sociodemographic and Farm-Level Factors That Predict Pig Farmers’ Self-Assessed Ability to Identify Pain in Pigs During Routine Husbandry Procedures

The sociodemographic and farm-level factors that predict pig farmers’ self-assessed ability to identify pain in pigs during routine husbandry procedure are shown in Table 4. The ordinal logistic regression conducted to determine the effect of age, gender, education level, years of experience, piggery size, and animal welfare awareness on farmers’ self-rated ability to assess pain in pigs revealed the overall model to be statistically significant, χ^2^ (10) = 32.803, *p* < 0.001, and met the proportional odds assumption (Test of Parallel Lines: *p* = 0.054). The size of the piggery (*p* = 0.014) and age group 34–41 (*p* = 0.025) were significant predictors. Farmers with larger piggeries were 1.77 times more likely (1/0.566) to rate their pain assessment ability higher. Farmers aged 34–41 were about 53.4% less likely to report high pain assessment ability compared to those aged > 41.

### 3.4. Comparison of Demographic Variables with Pain Perception

The nonparametric test of the comparison of demographic variables with pain perception is shown in Table 5. The aim of this analysis was to determine whether perceived pain intensity associated with routine husbandry procedures differs significantly across various groups of farmers based on their gender, age, education level, farming experience, piggery size, and self-assessed ability to recognise pain in animals. The comparison result showed that most of the comparison variables were not significant (*p* > 0.05). However, the perceived pain intensity associated with teeth clipping and years of experience was significant (*p* < 0.05). Likewise, the ability to assess pain by the respondents was significant (*p* < 0.05) when compared to vaccination pain perception.

### 3.5. Relationship Between Farmers Perceived Pain Intensity During Routine Husbandry Procedures and Their Knowledge, Welfare Awareness, Behavioural Observation Ability, and Related Attitudinal Factors

The correlation analysis between farmers perceived pain intensity during routine husbandry procedures and their knowledge, welfare awareness, behavioural observation ability, and related attitudinal factors is shown in Table 6. The ability to assess pain by the respondents was negatively correlated (r = −0.172, *p* = 0.016) with vaccination pain rating. Likewise, the knowledge of pain control inversely correlates with vaccination (r = −0.347, *p* = 0.000) and teeth clipping (r = −0.146, *p* = 0.041) pain ratings. Similarly, the correlation between castration and teeth clipping pain (r = 0.328, *p* = 0.000) ratings, and animal identification and vaccination (r = 0.437, *p* = 0.000) pain ratings were positively correlated. However, castration pain rating was inversely correlated with observation of behavioural change (r = −0.161, *p* = 0.024) and animal welfare awareness (r = −0.150, *p* = 0.035).

## 4. Discussion

A notable proportion of respondents agreed that farm animals benefit from pain alleviation, a finding likely linked to their education level and farming experience. Most participants had at least a high school education and extensive hands-on experience, which may enhance their awareness of animal welfare principles. As highlighted by Flecknell et al. [25], unmanaged pain can cause pathological changes in an animal’s physiology and behaviour, ultimately reducing quality of life and causing suffering. Such pain-related effects include sleep disruption, reduced feed intake, and diminished expressions of key behaviours such as grooming and play [26,27,28]. These deprivations can severely impact animal welfare [29]. At the same time, over half of the respondents agreed that the high cost of pain-relieving drugs prevents their regular use. This perception aligns with other commonly reported barriers, including mandatory withdrawal periods for drug residues, the limited availability of licensed analgesics for production animals, and the challenges of recognising and assessing pain in livestock [30,31].

The analysed data showed that educational level and years of experience were significantly associated with knowledge and perception related to animal welfare and pain. Participants with higher education had greater chances to accurately assess pain, highly rate their knowledge of pain control and allocate greater importance to animal welfare. According to Hemsworth et al. [22], education level is positively correlated with attitudes towards animal welfare and the utilization of pain mitigation strategies. It was noted that better-educated people are likely to be more empathetic toward animals and have enhanced observational abilities [22]. Previous studies examining attitudes toward pain and its mitigation in farm animals have also found that graduates gave higher scores when asked to quantify painful conditions [32]. Similarly, older farmers frequently have acquired experience that improves their sensitivity to subtle behavioural changes, even with no formal training. This is in line with the results, which revealed that age had a significant influence on aspects such as the ability to assess pain and understanding of pig behaviour [33]. However, gender did not significantly affect any knowledge or perception outcome; this suggests that gender may not inherently affect attitudes toward animal pain.

The findings clearly showed a significant association between the type of procedure (castration, tail docking, teeth clipping, and identification) conducted and the use of analgesics. The use of pain management techniques was strongly associated with painful practices like teeth clipping, tail docking, and castration. Hay et al. [34] and Windsor [35] support these findings, stating that farmers and veterinarians are more likely to administer analgesics during invasive treatments because they are more conscious of how unpleasant they are. However, as Weary et al. [36] also point out, the comparatively low usage of analgesics for procedures such as identification and vaccination may be due to an underestimation of the discomfort associated with these actions. The usage of analgesics was also significantly influenced by the person doing the procedure. This suggests that welfare methods vary according to personal accountability or degree of training.

The results of the ordinal logistic regression revealed that age and piggery size were significant predictors of farmers’ perceived ability to assess pain in pigs. Specifically, younger farmers were more likely to rate themselves as having a higher ability to recognize pain compared to their older counterparts. This finding aligns with earlier studies, which suggest that younger producers often demonstrate more progressive attitudes toward animal welfare and are more receptive to contemporary husbandry practices [37,38]. Younger individuals may also have greater exposure to recent training, education, or extension materials emphasizing animal sentience and the importance of pain mitigation. On the other hand, older farmers, who may rely more heavily on traditional practices, could be less familiar with pain indicators or the availability of analgesic interventions. These generational differences highlight the need for age-specific training programs aimed at enhancing pain assessment competencies across all farmer demographics.

Similarly, farmers managing larger piggeries were significantly more likely to report higher self-rated ability to assess pain in pigs. This trend likely reflects differences in operational scale, resource availability, and access to veterinary or technical support. Larger farms are often subject to higher market expectations and regulatory oversight, which can incentivize the adoption of welfare-enhancing practices such as the use of analgesics or routine pain monitoring [33,39]. These operations may also benefit from more structured staff training and investment in welfare-related infrastructure. In contrast, small-scale farms may face practical limitations, including limited personnel, budget constraints, or lack of access to veterinary services, that reduce their ability to implement or prioritize pain assessment protocols. Together, these findings underscore the importance of targeted extension services that consider both the scale of production and the demographic profile of pig farmers to improve welfare outcomes in extensive and intensive production systems alike.

It is noteworthy that education level, gender, and farming experience did not significantly predict farmers’ self-rated ability to assess pain in pigs. This finding contradicts the general assumption that higher education or longer farming experience enhances technical competence and welfare awareness. Previous studies have often associated formal education with better understanding of animal welfare principles and greater openness to adopting welfare-friendly practices [39,40]. However, the absence of such an effect in this study may suggest that educational attainment among pig farmers in the Eastern Cape is largely non-specialized and does not necessarily translate into practical knowledge of animal pain assessment. Similar observations have been reported in smallholder contexts, where formal education tends to be general rather than agriculture- or welfare-specific, limiting its direct impact on farm-level decision-making [23,41].

The lack of gender and experience effects further highlights that pain recognition may depend more on exposure to training, farm management scale, or access to veterinary guidance than on demographic attributes alone. Previous research indicates that long-term experience can reinforce traditional, routine practices rather than stimulate behavioural change, particularly when farmers have limited access to updated information or technical support [33,38]. Likewise, while gender differences in livestock management roles are well documented, their influence on welfare perception often varies with cultural norms and farm organization [23].

The correlation between pig farmers’ perceived pain intensity during routine husbandry procedures and various attitudinal, knowledge, and behavioural factors revealed several significant but weak-to-moderate associations, offering insight into the cognitive and experiential dimensions of animal welfare perception among on-farm workers. The correlation of pain assessment ability and vaccination revealed that farmers who rated their pain assessment ability higher were less likely to rate vaccination as highly painful. It may reflect confidence, bias or experience. This may suggest that farmers who perceive themselves as competent in identifying pain are more likely to downplay the pain caused by routine procedures. Similar observations have been made by [42], who noted that self-confidence in pain recognition may not always align with actual behavioural sensitivity to animal discomfort.

Knowledge of pain control was negatively correlated with pain ratings for vaccination and teeth clipping. This may reflect a desensitization effect among experienced or trained individuals who normalize certain painful procedures due to familiarity and repeated exposure. Literature has highlighted that increased exposure to invasive practices can sometimes dull empathetic responses over time [22,37]. On the other hand, a moderate positive correlation was observed between ratings of different painful procedures, such as castration and teeth clipping, and identification and vaccination. These results indicate internal consistency in pain perception across procedures, suggesting that farmers with higher overall sensitivity to animal discomfort tend to score all painful events similarly.

Contrary to expectations, awareness of animal welfare and importance attached to it were not significantly correlated with pain ratings, except for a weak negative association between castration pain and welfare awareness status. This could imply that welfare awareness alone may not influence pain sensitivity unless it is accompanied by hands-on training or active engagement in pain mitigation practices. This aligns with findings by Sinclair et al. [43], who emphasized that awareness does not always translate into compassionate practice unless reinforced by institutional policies or economic incentives. A final observation was the weak negative correlation between observed behavioral changes and perceived castration pain. This may suggest a disconnect between objective animal responses and subjective human interpretations, supporting earlier work by [44] indicating that reliance on behaviour alone can underestimate true pain experiences in livestock.

The observed gaps in pain recognition and inconsistent use of analgesics therefore extend beyond animal welfare concerns, influencing physiological stress responses that ultimately affect meat quality and market value. Inadequate pain management and welfare compromise during routine husbandry procedures have tangible implications for pig meat quality. Painful or poorly managed interventions such as castration, tail docking, and teeth clipping can trigger prolonged physiological stress responses characterized by elevated cortisol, catecholamine secretion and other stress-induced biochemical changes, ultimately leading to meat quality defects [45]. Thus, addressing pain management at the farm level is central to both sustainable welfare compliance and improved product quality outcomes.

## 5. Conclusions

This study highlights important gaps in the perception and management of pain among pig farmers in the Eastern Cape, South Africa. Although most respondents acknowledged the benefits of pain alleviation, effective implementation of analgesic use during routine husbandry practices such as castration, tail docking, and teeth clipping remained limited. Farmers’ education level, years of farming experience, and piggery size were key factors influencing welfare awareness and knowledge of pain control, yet these did not necessarily predict higher practical competence in pain assessment. The ordinal regression results revealed that younger farmers and those managing larger piggeries were more likely to rate themselves as having a higher ability to recognize pain, suggesting that generational and structural factors play stronger roles than formal education or gender.

However, the negative correlations between self-rated pain assessment ability and perceived pain intensity for routine procedures suggest a possible overconfidence bias, where farmers who consider themselves skilled may underestimate the severity of pain experienced by pigs. This disconnect underscores the need for evidence-based training focused on objective pain indicators and behavioural assessment.

Overall, improving farmer education, age-targeted extension programs, and veterinary outreach are crucial to strengthening on-farm welfare practices. Integrating welfare-based training modules into agricultural extension and enforcing standardized pain management guidelines could promote humane pig production and enhance animal well-being.

## Figures and Tables

**Table 1 animals-15-03508-t001:** Sociodemographic characteristics of respondents.

Variables	Groups	Percentage (%)
Age	18–25	10.2
	26–33	20.4
	34–41	37.2
	>41	32.1
Gender	Male	74.0
	Female	26.0
Marital status	Married	39.3
	Divorced	3.6
	Single	50.5
	Widowed	5.6
	Separated	1.0
Household number	<5	42.9
	5–10	49.0
	>10	8.2
Monthly income	<5000	18.4
	5000–10,000	46.9
	>10,000	34.7
Income source	Salaries	40.3
	Pension/social grant	3.1
	Livestock and Crop	54.6
	Others	4
Employment status	Employed	44.4
	Unemployed	5.6
	Private business	45.4
	Other	4.6
Highest qualification	Primary	7.1
	High school	28.6
	Diploma/technical college	24.5
	University degree	37.8
	None	2.0
Religion	Christianity	66.3
	Islam	0.5
	Traditionalist	32.7
	Atheist	0.5
Size of piggery farm	Small (1–100)	52.0
	Medium (101–500)	38.3
	Large (>500)	9.7
Purpose	Meat production	55.1
	Breeding	3.1
	Sales of piglets	20.4
	Manure production	0.5
	Research Purpose	6.6
	Income generation	14.3
Ability to assess pain in pig	Excellent	15.3
	Very good	26.5
	Good	33.2
	Fair	22.4
	Very poor	2.6
Knowledge of pain control in pig	Excellent	19.4
	Very good	33.2
	Good	28.6
	Fair	17.3
	Poor	1
	Very poor	0.5

**Table 2 animals-15-03508-t002:** Perception about pain management during routine husbandry procedures.

Variable	Strongly Agree %	Agree %	Neither Agree nor Disagree %	Disagree %	Strongly Disagree %
Farm animals benefit from pain alleviation	21.4	30.1	8.7	25.0	14.8
The current management of animals at my farm offers sufficient opportunity to identify animals in pain	27.6	54.1	7.1	9.7	1.5
Pain relieving drugs are too expensive to use regularly	17.3	44.9	19.4	14.8	3.6
Providing pain relief is impracticable most of the time as a result of the need for increased time and labour	11.7	39.8	16.3	26.5	5.6
Difficulties with gathering and/or handling means that it is very difficult to administer pain relief	9.7	52.6	16.3	17.3	4.1
Pain reliving drugs are not necessary for farm animals	3.6	7.1	11.2	43.9	34.2
So much cost is involved	20.9	52.0	17.3	7.1	2.6
Animals recover better after administering drug	36.7	46.9	10.7	3.6	2.0
It is difficult to recognize pain in farm animals	3.6	11.7	14.3	49.5	20.9
Some degrees of pain are beneficial to the animal	8.7	17.3	17.9	28.6	27.6

**Table 3 animals-15-03508-t003:** Chi-square association analysis of selected variables.

Factors	χ^2^	df	*p*-Values
A. Socio-demographics vs. Knowledge/Practices/Perception			
Gender vs. knowledge of pain control	3.145	5	0.678
Education level vs. awareness of animal welfare	2.312	4	0.679
Education level vs. ability to assess pain	32.50	20	0.038
Education level vs. knowledge rating of pain control	65.32	20	0.000
Education level vs. importance of animal welfare	36.45	20	0.014
Education level vs. ability to identify an animal in pain	39.51	16	0.001
Age vs. ability to assess pain	25.05	15	0.049
Age vs. knowledge rating of pain control	19.021	15	0.213
Age vs. importance of animal welfare	22.01	15	0.107
Age vs. ability to identify animals in pain	13.84	12	0.311
Age vs. understanding of behaviour in pigs	3.41	3	0.333
Age vs. knowledge of behavioural expression in pigs	8.59	3	0.035
Age vs. behavioural observation in pigs	4.35	3	0.226
Years of experience vs. ability to assess pain	19.08	10	0.039
Years of experience vs. knowledge rating of pain control	11.91	10	0.291
B. Routine husbandry procedure vs. Pain Control Practices			
Procedure performed (yes/no) vs. use of analgesics for the procedure (identification)	0.691	1	0.406
Procedure performed (yes/no) vs. use of analgesics for the procedure (vaccination)	2.430	1	0.119
Procedure performed (yes/no) vs. use of analgesics for the procedure (castration)	183.46	2	0.000
Procedure performed (yes/no) vs. use of analgesics for the procedure (teeth-clipping)	21.538	2	0.000
Procedure performed (yes/no) vs. use of analgesics for the procedure (tail-docking)	21.570	2	0.000
Person responsible vs. use of analgesics (identification)	53.54	3	0.000
Person responsible vs. use of analgesics (vaccination)	32.22	6	0.121
Person responsible vs. use of analgesics (castration)	50.09	6	0.000
Person responsible vs. use of analgesics (tail-docking)	51.90	6	0.000
Person responsible vs. use of analgesics (teeth-clipping)	32.87	6	0.000
C. Behavioural Awareness vs. Perception			
Education level vs. understanding of animal behaviour	5.05	4	0.282
Education level vs. knowledge of behavioural expression	1.319	4	0.858
Education level vs. behavioural observation during routine husbandry procedure	11.28	4	0.024
Knowledge of normal pig behaviour vs. observatory frequency of abnormal behaviour	0.195	1	0.658
Gender vs. perceived pain intensity during castration	4.295	3	0.231
Gender vs. perceived pain intensity during teeth-clipping	2.483	4	0.648
Gender vs. perceived pain intensity during vaccination	3.890	4	0.421
Gender vs. perceived pain intensity during identification	3.023	4	0.554
Years of experience vs. perceived pain intensity during castration	9.639	6	0.141
Years of experience vs. perceived pain intensity during teeth-clipping	15.599	8	0.048
Years of experience vs. perceived pain intensity during vaccination	7.860	8	0.447
Years of experience vs. perceived pain intensity during identification	10.562	8	0.228
Knowledge of behaviour vs. recognition of behavioural change post-procedure	0.064	1	0.801
Years of experience vs. ability to identify pain	18.77	8	0.016

**Table 4 animals-15-03508-t004:** Ordinal logistic regression predicting farmers’ ability to assess pain in pigs.

Predictor	B (Estimate)	Std. Error	Wald	*p*-Value	95% CI (Lower, Upper)	Exp (B)
Size of piggery	−0.568	0.230	6.101	0.014	[−1.020, −0.117]	0.566 ^s^
Age;						
Age (34–41)	−0.765	0.342	5.013	0.025	[−1.434, −0.095]	0.466 ^s^
Age (18–25)	−0.289	0.528	0.299	0.584	[−1.325, 0.747]	0.749 ^ns^
Age (26–33)	0.229	0.437	0.276	0.600	[−0.627, 1.085]	1.258 ^ns^
Educational Qualification	−0.137	0.141	0.945	0.331	[−0.414, 0.140]	0.872 ^ns^
Years of farming experience	−0.329	0.209	2.491	0.115	[−0.738, 0.080]	0.720 ^ns^
Gender (Male)	−0.562	0.306	3.370	0.066	[−1.162, 0.038]	0.570 ^ns^
Type of piggery; conventional	−0.098	0.319	0.095	0.758	[−0.724, 0.527]	0.907 ^ns^
organic	−0.001	0.351	0.000	0.998	[−0.689, 0.687]	0.999 ^ns^
Animal Welfare	−0.895	0.533	2.824	0.093	[−1.940, 0.149]	0.409 ^ns^

s = significant; ns = not significant; CI = confidence interval. Reference category: Gender (female); Age group (>41 years); Educational qualification (University/tertiary); Years of faming experience (Most experienced group: >10 years); Size of piggery (Largest piggery size: >100 pigs); Animal welfare awareness (very poor).

**Table 5 animals-15-03508-t005:** Nonparametric test of demographic variables comparison with pain perception.

Procedure	Grouping Variable	Test	Chi-Square/U	df	*p*-Value
Castration	Gender	Mann–Whitney U	3596.5	—	0.747
Identification			3251.5	—	0.167
Vaccination			3459.0	—	0.423
Teeth Clipping			3236.0	—	0.139
Castration	Educational level	Kruskal–Wallis	4.628	4	0.328
Identification			2.642	4	0.619
Vaccination			7.168	4	0.127
Teeth Clipping			2.688	4	0.611
Castration	Years of experience	Kruskal–Wallis	2.888	2	0.236
Teeth Clipping			6.256	2	0.044
Vaccination			0.682	2	0.711
Identification			0.443	2	0.801
Castration	Age	Kruskal–Wallis	1.595	3	0.661
Identification			0.561	3	0.905
Vaccination			2.287	3	0.515
Teeth Clipping			1.777	3	0.620
Castration	Piggery Size	Kruskal–Wallis	2.088	2	0.352
Identification			1.079	2	0.583
Vaccination			1.701	2	0.427
Teeth Clipping			2.965	2	0.227
Identification	Ability to assess pain	Kruskal–Wallis	3.925	4	0.416
Vaccination			13.759	4	0.008
Teeth Clipping			1.634	4	0.803
Castration			2.490	4	0.646

**Table 6 animals-15-03508-t006:** Correlation Analysis between farmers perceived pain intensity during routine husbandry procedures and their knowledge, welfare awareness, behavioural observation ability, and related attitudinal factors.

Variable 1	Variable 2	Correlation (r)	*p*-Value
A. Pain perception vs. knowledge/ability attributes			
Ability to assess pain	Animal identification pain rating	−0.021	0.771 ^ns^
Ability to assess pain	Vaccination pain rating	−0.172	0.016 ^s^
Ability to assess pain	Teeth clipping pain rating	−0.036	0.618 ^ns^
Ability to assess pain	Castration pain rating	−0.100	0.162 ^ns^
Castration pain rating	Knowledge of pain control	−0.046	0.523 ^ns^
Teeth clipping pain rating	Knowledge of pain control	−0.146	0.041 ^s^
Vaccination pain rating	Knowledge of pain control	−0.347	0.000 ^s^
Identification pain rating	Knowledge of pain control	−0.094	0.190 ^ns^
Identification pain rating	Importance of welfare	−0.001	0.988 ^ns^
Vaccination pain rating	Importance of welfare	0.023	0.752 ^ns^
Teeth clipping pain rating	Importance of welfare	−0.021	0.768 ^ns^
Castration pain rating	Importance of welfare	0.001	0.992 ^ns^
B. Inter-pain rating correlations			
Castration pain rating	Teeth clipping pain rating	0.328	0.000 ^s^
Identification pain rating	Vaccination pain rating	0.437	0.000 ^s^
C. Welfare awareness vs. Pain rating			
Identification pain rating	Behavioural change observation	−0.018	0.800 ^ns^
Vaccination pain rating	Behavioural change observation	0.029	0.686 ^ns^
Teeth clipping pain	Behavioural change observation	0.064	0.370 ^ns^
Castration pain rating	Behavioural change observation	−0.161	0.024 ^s^
Identification pain rating	Welfare awareness	0.028	0.693 ^ns^
Vaccination pain rating	Welfare awareness	0.023	0.744 ^ns^
Teeth clipping pain	Welfare awareness	−0.042	0.556 ^ns^
Castration pain rating	Welfare awareness	−0.150	0.035 ^s^

s = significant; ns = not significant.

## Data Availability

The original contributions presented in this study are included in the article. Further inquiries can be directed to the corresponding author.

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
