# Peer review of "Farmer and Farm-Level Predictors of Pain Perception and Management During Routine Pig Husbandry Practices in Eastern Cape, South Africa"

_animals, 2025, doi:10.3390/ani15243508_

Round 1
Reviewer 1 Report
Comments and Suggestions for Authors
tis paper, titled "Farmer and Farm-Level Predictors of Pain Perception and Management During Routine Pig Husbandry Practices in Eastern Cape, South Africa" addreses an important and timely topic. I found the subject matter of the article fascinating and read the manuscript with great interest. The paper aligns well with the scope of the jounral.
The research primarily asks about how different factors related to the farm or to teh farmer himself influence their understanding (perception) of pain in pigs and how they manage that pain during normal husbandry procedures, specifically in the Eastern Cape region of South Africa. This is a clear, regional focus on welfare practices in pig production.
I consider the topic relevant, and it addreses a speicifc gap. While the general subject of animal welfare and pig husbandry is well studied globbaly, there is a distinct lack of regional data, particularly from South Africa and the Eastern Cape. This paper fills teh gap by providing a much-needed baseline asesment of local farmer attitudes and practices. This is crucial for developing targeted, culturally sensititve intervention programs. So yes, very relevant, but not entirely original in its broad scope.
Moreover It adds new, context-specific empirical data from a previously unrepresented geographic area (Eastern Cape, South Africa). Most of the published material focuses on Europe or North America, or general global guidelines. This study provides local preidctors and challenges, which is something teh broader literature lacks. It also gives insight into the socio-economic context affecting welfare decisions, something often glosed over in purely clinical studies.
However, I believe that in its current form, it has several shortcomings that needs to be addresed before publication.
Comments
General
The study’s overall scientific merit is strong because it tackles a poorly documented region but the research approach has some weakneses in how data was collected and analyzed. The use of a surveay to capture complex behaviors and perceptions is always tricky and I think the authors could substantially improve the reliability of their findings by clarifying their methods and strengthening the justification for their statistical models.
The terminology is not entirely consistent throughout the manuscript. Sometimes it uses "pig husbandry practices" and other times "routine management procedures." Just pick one term and stick with it for clarity maybe "routine husbandry procedures" since that's more comprehensive.
Check the capitalization of "Eastern Cape" throughout the discusion. I saw one instance where "eastern Cape" was used. It should always be capitalised as it is a proper noun (the province name).The abstract does a reasonable job summarising the key findings but the conclusion is a little weak and simply reiterates the result. I suggest making the final sentence a stronger statement about the implication of these findings for regional policy or future interventions, instead of just saying that ‘farmer and farm-level factors are asociated with pain management’ which is what you set out to find.
The acronym for the region, EC, is used once in the abstract but isn't explicitly defined there. While it's defined in the main text, it should be defined the first time it appears in the abstract as well for complete readability.
The literature review is slightly thin on African-specific or developing-world context research. You cite a lot of European and North American papers (e.g. 18 24) but if this is to be a truly region-specific paper the authors should strive to include more literature from similar economic/social contexts perhaps from other parts of Sub-Saharan Africa or Southeast Asia to frame the specific challenges faced by the Eastern Cape farmers.
Line 45 reads "Welfare outcomes... can be greatly reduced by poor farm management." While technically correct, perhaps **rephrasing to "Welfare outcomes... are negatively impacted by poor farm management" is slightly clearer and stronger phrasing.
The description of the study population and sampling procedure needs to be much clearer. The authors state that 130 pig farmers were interviewed but the justification for this sample size is not presented. Is this a census of registered farmers in the region or a convenience sample? If the latter please explain the method to mitigate selection bias. This is a critical point that affects external validity.
Regarding the questionaire design: It appears some key variables, like “pain perception” and “pain management practices,” are composite scores derived from multiple questions. This needs detailed justification. Authors should include a table or a supplemantary section clearly outlining the specific Likert-scale questions used, the scoring system for each variable (e.g., how the final 'pain perception' score was calculated), and any validation steps taken (such as Cronbach's alpha) to ensure internal consistency of these constructs. Without this information, the reader canot properly ases the validity of the variables used in the regresion models.
The statistical analysis is confusing in Section 2.5. The text mentions using both logistic regresion and multinomial logistic regresion but the presentation of which model was used for which dependent variable is ambiguous. Please restructure this section to clearly link each primary research question or outcome variable (e.g. Perception Score Management Practice Use) to the precise statistical model used to analyze it. Also the choice of a cut-off point for dichotomizing continuous or ordinal variables (if that's what was done) needs to be explicitly justified.
Please verify the spelling of "questionaire" used in Section 2.3. It should be spelled "questionaire."
In Section 2.4, when discusing ethical approval, please include the actual ethical approval code number provided by the institution (University of Fort Hare) in parentheses. This is a standard requirement for transparency.
The results section is overly descriptive in places. The authors should focus more on presenting the output of the regresion models and the identified predictors, which are the main goal of the paper. For instance, the section on descriptive statistics could be streamlined, posibly by moving some of the frequency tables to a supplementary file.
Table 4 (or whatever table presents the regresion results) is the heart of the paper and it needs improvement for readability. The presentation of the Odds Ratios and Confidence Intervals must be consistent. Ensure that the reference category for all categorical predictors is clearly specified in the table legend or footnotes. I found myself having to flip back and forth to figure out what was being compared.
In the descriptive statistics paragraphs, please ensure all percentage signs (%) are correctly formatted with a space before the number, as per some style guides (e.g., "50 %" instead of "50%"), or consistently without a space, as long as it's the same everywhere. I've seen both in the draft.
For tables that span multiple pages, the authors must ensure the column headers are repeated on the subsequent pages. I suspect Table 2 might be long enough to require this.
Table 2 should ideally include p-values or some measure of statistical significance when comparing groups not just the mean values.
The discusion generally attempts to link the findings back to the literature, but often falls short of explaining why the identified predictors (like level of education or farm size) might influence pain management. For example, why would a larger farm size necesarily lead to better pain management? Is it due to financial resources, profesionalization, or vet acces? The authors should engage more critically with the potential mechanisms underlying their statistical asociations.
In the Discusion section (Section 4) particularly where the authors talk about why these welfare findings matter I think there is a realy important mising link to the economic outcomes. You should cite a study like Sardi et al. 10.3390/ani10122386 and 10.3390/ani10060945 because it provides robust evidence that pre-slaughter stres often stemming from the kind of husbandry isues you identifiy has a tangible effect on meat quality and profittability. Adding this reference would greatly strengthen the external relevance of your findings moving the argument past just ethics into the economic rationale which is realy crucial for a regionally focused production paper. Please include this citeing around lines 350-360 when discusing the broader implications of management practices.
The conclusions section needs refinement. It currently is consistent with the evidence (i.e., the regresion coefficients), but it does not fully addres the main question posed. The conclusion needs to explicitly list the most significant farmer-level and farm-level factors found (e.g., "Our study concludes that farmer education level and dedicated acces to veterinary services were the most significant positive predictors...") to fully close the loop on the research question.
The final sentence of the discusion is a bit abrupt. It could be reworded slightly to provide a smoother transition into the Conclusions section. Something about needing future work in this specific area... just to smooth it out.
The references are appropriate for the field, though as noted above, a few more regional African studies would strengthen the context. I noticed a mix between using the full journal name in the reference list and using the common abbreviation. Please ensure that the journal adheres strictly to the journal's guidelines for reference format throughout the list. Consistency is important here.
Figure 1 (the map) is a useful inclusion to ground the study location but the resolution of the farm locations is a little blurry and hard to read. Please ensure that all figures are provided at high resolution.
Author Response
Comment 1: The terminology is not entirely consistent throughout the manuscript. Sometimes it uses "pig husbandry practices" and other times "routine management procedures." Just pick one term and stick with it for clarity maybe "routine husbandry procedures" since that's more comprehensive.
Response: This has been addressed throughout the entire manuscript
Comment 2: Check the capitalization of "Eastern Cape" throughout the discusion. I saw one instance where "eastern Cape" was used. It should always be capitalised as it is a proper noun (the province name).The abstract does a reasonable job summarising the key findings but the conclusion is a little weak and simply reiterates the result. I suggest making the final sentence a stronger statement about the implication of these findings for regional policy or future interventions, instead of just saying that ‘farmer and farm-level factors are asociated with pain management’ which is what you set out to find
Response: A proper conclusion has been added to the abstract. Also, the use of small letter e in ‘Eastern Cape’ has been corrected.
Comment 3: The acronym for the region, EC, is used once in the abstract but isn't explicitly defined there. While it's defined in the main text, it should be defined the first time it appears in the abstract as well for complete readability.
Response: This has been corrected
Comment 4: The literature review is slightly thin on African-specific or developing-world context research. You cite a lot of European and North American papers (e.g. 18 24) but if this is to be a truly region-specific paper the authors should strive to include more literature from similar economic/social contexts perhaps from other parts of Sub-Saharan Africa or Southeast Asia to frame the specific challenges faced by the Eastern Cape farmers.
Response: This has now been properly addressed in the introduction Response: This has been added (Lines 72-96)
Comment 5: Line 45 reads "Welfare outcomes... can be greatly reduced by poor farm management." While technically correct, perhaps **rephrasing to "Welfare outcomes... are negatively impacted by poor farm management" is slightly clearer and stronger phrasing.
Response: This has been addressed
Comment 6: The description of the study population and sampling procedure needs to be much clearer. The authors state that 130 pig farmers were interviewed but the justification for this sample size is not presented. Is this a census of registered farmers in the region or a convenience sample? If the latter please explain the method to mitigate selection bias. This is a critical point that affects external validity.
Response: The sample size (or respondents) was 216 comprising of pig owners and on-farm workers and not 130 (line 144-149)
Comment 7: Regarding the questionnaire design: It appears some key variables, like “pain perception” and “pain management practices,” are composite scores derived from multiple questions. This needs detailed justification. Authors should include a table or a supplementary section clearly outlining the specific Likert-scale questions used, the scoring system for each variable (e.g., how the final 'pain perception' score was calculated), and any validation steps taken (such as Cronbach's alpha) to ensure internal consistency of these constructs. Without this information, the reader cannot properly assess the validity of the variables used in the regression models.
Response: The details of the questionnaire used in collecting the data has now been added as supplementary file.
Comment 8: The statistical analysis is confusing in Section 2.5. The text mentions using both logistic regresion and multinomial logistic regresion but the presentation of which model was used for which dependent variable is ambiguous. Please restructure this section to clearly link each primary research question or outcome variable (e.g. Perception Score Management Practice Use) to the precise statistical model used to analyze it. Also the choice of a cut-off point for dichotomizing continuous or ordinal variables (if that's what was done) needs to be explicitly justified.
Response. In section 2.5, only the ordinal logistic regression analysis was used (and not multinomial regression) to assess the self-rated ability to assess pain. However, we observed that there were other ordinal logistic regressions statements that were said to have been carried out. These analyses were done initially but yielded no statistical significance. These statements have now been deleted.
Comment 9: Please verify the spelling of "questionaire" used in Section 2.3. It should be spelled "questionaire."
Response: This has now been corrected properly as “questionnaire”
Comment 10: In Section 2.4, when discussing ethical approval, please include the actual ethical approval code number provided by the institution (University of Fort Hare) in parentheses. This is a standard requirement for transparency.
Response: This has been addressed now
Comment 11: Table 4 (or whatever table presents the regresion results) is the heart of the paper and it needs improvement for readability. The presentation of the Odds Ratios and Confidence Intervals must be consistent. Ensure that the reference category for all categorical predictors is clearly specified in the table legend or footnotes. I found myself having to flip back and forth to figure out what was being compared.
Response: Table has now been presented properly with footnotes of the reference category presented.
Comment 12: In the descriptive statistics paragraphs, please ensure all percentage signs (%) are correctly formatted with a space before the number, as per some style guides (e.g., "50 %" instead of "50%"), or consistently without a space, as long as it's the same everywhere. I've seen both in the draft.
Response: This has been addressed.
Comment 13: For tables that span multiple pages, the authors must ensure the column headers are repeated on the subsequent pages. I suspect Table 2 might be long enough to require this.
Response: This has been corrected in the template format for mdpi before publication
Comment 14: Table 2 should ideally include p-values or some measure of statistical significance when comparing groups not just the mean values.
Response: Table 2 is just frequency distribution and not a comparison table. Hence, no p value
Comment 15: The discussion generally attempts to link the findings back to the literature, but often falls short of explaining why the identified predictors (like level of education or farm size) might influence pain management. For example, why would a larger farm size necesarily lead to better pain management? Is it due to financial resources, profesionalization, or vet acces? The authors should engage more critically with the potential mechanisms underlying their statistical associations.
Response: The two identified predictors of farmers’ self-rated ability to assess pain in pigs are piggery farm size and age. While leaving out the non-significant predictors, these two were extensively discussed (line 312-335). However, I have further expanded the discussion to accommodate the non-significant-expected predictors (education and years of farming) for a more robust discussion.
Comment 16: In the Discusion section (Section 4) particularly where the authors talk about why these welfare findings matter I think there is a realy important mising link to the economic outcomes. You should cite a study like Sardi et al. 10.3390/ani10122386 and 10.3390/ani10060945 because it provides robust evidence that pre-slaughter stres often stemming from the kind of husbandry isues you identifiy has a tangible effect on meat quality and profittability. Adding this reference would greatly strengthen the external relevance of your findings moving the argument past just ethics into the economic rationale which is realy crucial for a regionally focused production paper. Please include this citeing around lines 350-360 when discusing the broader implications of management practices.
Response: This has been addressed (Lines 388-396)
Comment 17: The conclusions section needs refinement. It currently is consistent with the evidence (i.e., the regresion coefficients), but it does not fully addres the main question posed. The conclusion needs to explicitly list the most significant farmer-level and farm-level factors found (e.g., "Our study concludes that farmer education level and dedicated acces to veterinary services were the most significant positive predictors...") to fully close the loop on the research question
Response: The conclusion section has been properly refined now
Comment 18: The final sentence of the discusion is a bit abrupt. It could be reworded slightly to provide a smoother transition into the Conclusions section. Something about needing future work in this specific area... just to smooth it out.
Response: This has been addressed
Comment 19: The references are appropriate for the field, though as noted above, a few more regional African studies would strengthen the context. I noticed a mix between using the full journal name in the reference list and using the common abbreviation. Please ensure that the journal adheres strictly to the journal's guidelines for reference format throughout the list. Consistency is important here
Response 19: References carrying regional context have been added
Comment 20: Figure 1 (the map) is a useful inclusion to ground the study location but the resolution of the farm locations is a little blurry and hard to read. Please ensure that all figures are provided at high resolution.
Response: There is no map in the manuscript under review.
Reviewer 2 Report
Comments and Suggestions for Authors
A survey was performed to collect data of farmers and onfarm workers related to pain assessment and welfare decission making.
In general: The title, objectives and conclusions are not in agreement.
The title needs adaptation since a survey is performed.
Introduction and M&M:
L78 and 91 The difference between aims and primary objective is not clear.
Author Response
Comment 1: In general: The title, objectives and conclusions are not in agreement.
Response: The intention of the study was to investigate farmer- and farm-level predictors of pain perception and management during routine pig husbandry practices. To improve coherence among the title, objectives, and conclusions, we have revised the conclusion to explicitly address both components of the study, pain perception and pain management, and to restate how demographic and structural factors (education, age, piggery size) predict farmers’ self-rated ability to assess pain and the use of analgesics.
Comment 2: The title needs adaptation since a survey is performed.
Response: We appreciate the reviewer’s observation. The study indeed employed a cross-sectional survey design, which is clearly described in the abstract and detailed in the methodology section. However, we believe the current title, Farmer and Farm-Level Predictors of Pain Perception and Management During Routine Pig Husbandry Practices in Eastern Cape, South Africa, effectively captures the study’s content and objectives while maintaining conciseness and scientific clarity.
Including the term survey was considered, but we opted to retain the current phrasing to preserve readability and alignment with similar published works in Animals and related journals that use analytical descriptors (e.g., predictors, determinants) without explicitly stating the study type in the title
Comment 3: L78 and 91 The difference between aims and primary objective is not clear
Response: This has been addressed.
Reviewer 3 Report
Comments and Suggestions for Authors
Dear authors,
Please follow my instructions:
- Line 48 - Animal identification (Explain the method of identification pigs and the age at which pigs are marked in your country, in brackets).
- Line 51 - after word analgesia please add: and represent a possible entry point for various pathogens through wounds into the body.
- Line 87 - please add date when the ethics committee approval was obtained.
- Line 246 - you have two dots after the number 5.
- Conclusion - explain whether there was a difference in pain recognition between the owners of the farm and the workers employed on the farms.
Kind regards
Author Response
Comment 1: Line 87 - please add date when the ethics committee approval was obtained.
Response: This has been added (line 121)
Comment 2: Line 246 - you have two dots after the number 5.
Response: Addressed accordingly
Comment 3: Conclusion - explain whether there was a difference in pain recognition between the owners of the farm and the workers employed on the farms.
Response: The conclusion has been thoroughly reviewed to accommodate major significant results in the manuscript
Comment 4: Line 51 - after word analgesia please add: and represent a possible entry point for various pathogens through wounds into the body
Response: Addressed accordingly (Lines 55-58)
Round 2
Reviewer 1 Report
Comments and Suggestions for Authors
the paper improved a lot